

# Sampling strategies for digestive system flora studies: current research and perspectives

Xiaobo Liu[1,2,*], Xia Cheng[3,*], Ziye Gao[4], Jun Pan[1], Shizhen Luo[1], Pei Liu[5], Hui Wen[1] and Shu Jin[1]

[1] Department of Gastroenterology, Taihe Hospital, Hubei University of Medicine, Shiyan, Hubei, China
[2] Hubei Provincial Clinical Research Center for Precision Diagnosis and Treatment of Liver Cancer, Shiyan, Hubei, China
[3] Department of Gastroenterology, Tongren People's Hospital, Tongren, Guizhou, China
[4] Department of Oncology, Taihe Hospital, Hubei University of Medicine, Shiyan, Hubei, China
[5] Department of Dermatology, Taihe Hospital, Hubei University of Medicine, Shiyan, Hubei, China
[*] These authors contributed equally to this work.

## ABSTRACT

**Background**. The digestive system is significantly associated with the incidence and progression of various diseases. Investigating its microbial community will aid in the diagnosis, treatment, and prognostication of digestive pathologies. Microbial composition varies not only between distinct organs but also within different regions of the same organ. Additionally, dynamic shifts in microbial composition occur throughout various phases of the disease, complicating research. This study systematically evaluated the biological samples and diverse collection methods employed in digestive system microbiota research, with the aim of guiding sample selection and collection strategies for future microbial studies.

**Methodology**. We conducted a comprehensive literature review using professional databases such as PubMed and Web of Science. The focus of this review is on microbial community research, particularly the use of high-throughput sequencing to compare different samples of the same organization, as well as the impact of different sampling methods on microbial community structure of the same sample.

**Results**. A diverse array of microbiological samples are available for study, including saliva, endoscopic biopsies, luminal mucosa, luminal fluid, feces, bile, and surgically excised tissues. Multiple techniques exist to obtain specimens from identical locations, each with a unique set of benefits and limitations.

**Conclusions**. When selecting specimens and sampling methods for microbiota studies, it is essential to consider the characteristics of the microbiota population, research environment, and the objectives of the study. Despite the importance of appropriate sampling for microbial community studies, consensus on the optimal sample type and collection method remains elusive.

Corresponding authors
Hui Wen, iriswenhui@163.com
Shu Jin, jinshu76@sohu.com

## INTRODUCTION

The human microbiota includes trillions of bacteria, archaea, fungi, and viruses, which are distributed across the skin, respiratory tract, oral cavity, and digestive tract (*Sender, Fuchs & Milo, 2016*). More than 70% of the human microbiota is found in the gastrointestinal tract, comprising thousands of microorganisms with a total weight of approximately 1.5 kg (*Kwon et al., 2021*). The establishment of the human microbiota commences post-birth, and by the age of approximately three years, the composition and diversity of the digestive tract microbiota closely resembles those of adults (*Rinninella et al., 2019*). The gastrointestinal tract microbiota is associated with age, race, dietary patterns, environmental factors, medication use, and health status and undergoes dynamic changes over time (*Wong, Harris & Ferguson, 2016*; *Ramirez et al., 2020*; *Zhang et al., 2023*). There are variations in microbiota composition among individuals.

The gastrointestinal tract microbiota actively participates in various physiological processes within the human body, including material metabolism, nutrient absorption, maintenance of the gastrointestinal mucosal barrier function, immune regulation, and prevention of pathogenic infections (*Jandhyala et al., 2015*). An imbalance in this microbiota can increase the risk of gastrointestinal and systemic diseases (*Adhikary et al., 2024*; *Zhu et al., 2023*). Conversely, diseases can also influence the composition of the digestive tract microbiota (*Lynge & Belstrom, 2019*). Traditional microbiological identification methods primarily rely on cultivation, which is constrained by the inability to culture certain digestive tract microbiotas (*Almeida et al., 2019*). Advanced sequencing technologies, such as whole-genome shotgun and 16S ribosomal RNA (16SrRNA) gene analysis, have revolutionized microbiota research by accurately characterizing the digestive tract microbiota without necessitating culture.

Investigation of the gut microbiota holds significant potential for aiding in the diagnosis, treatment, and prognosis of diseases. Various sources, including endoscopic biopsy tissue, mucosa of the intestinal wall, intraluminal fluid, gastrointestinal secretions, excretions, bile, and surgical resection tissue, can serve as viable materials for studying gastrointestinal microbiota. In oral microbiota research, samples can be obtained through saliva collection, oral mucosal samples, tissue biopsies, and dental plaque. For the esophagus, stomach, small intestine, and colorectal regions, endoscopic brushings and biopsy tissues are commonly used sampling methods. Additionally, luminal fluids, gastrointestinal secretions, and excreta can be utilized for microbiota studies in these anatomical areas. Specifically, rectal swabs serve as a targeted approach for investigating colorectal microbial communities. Liver specimens can be obtained through surgical resection, percutaneous liver biopsy, or laparoscopic biopsy, while noninvasive methods include blood, stool, and bile specimen collection. Pancreatic sample acquisition is often challenging and may involve surgical procedures, endoscopic ultrasound-guided fine-needle aspiration (EUS-FNA), or endoscopic retrograde cholangiopancreatography (ERCP). Furthermore, blood and stool specimens can be utilized for pancreas-related research. Notably, the microbial composition of samples obtained from different sources and through distinct sampling methods may exhibit discrepancies. Hence, the selection of appropriate samples and

sampling methodologies is critical for reliable microbial community research. This study systematically assessed digestive tract microbial samples and sampling techniques with the aim of offering valuable insights for future research on the digestive tract microbiota and the rational selection of sampling methods.

## METHODS

### Search strategy

The systematic review was conducted by following Preferred Reporting Items for Systematic reviews, and Meta-Analyses (PRISMA) statement (*Moher et al., 2009*). All data were based on published studies, and no ethical issues were involved. Relevant studies on the gastrointestinal tract and microbiota relationship that were published from a certain date until April 2025 were retrieved from Pubmed, EMbase, Scopus, Cochrane Library, Embase, Wiley Online Libraryne. We attempted to trace the references that had been incorporated into literature and manually retrieve the relevant conference proceedings to identify potential information that had not been retrieved.

The search strategy used was as follows: ((((((((((((((((((((((Mouth) OR (Cavitas Oris)) OR (Oral Cavity)) OR (Cavity, Oral)) OR (Vestibule of the Mouth)) OR (Vestibule Oris)) OR (Oral Cavity Proper)) OR (Cavitas oris propria)) OR (Mouth Cavity Proper)) OR (Esophagus)) OR ((Stomach) OR (Stomachs))) OR ((Duodenum) OR (Duodenums))) OR ((Jejunum) OR (Jejunums))) OR (Ileum)) OR (((((((Colon) OR (Appendix Epiploica)) OR (Omental Appendices)) OR (Appendices, Omental)) OR (Omental Appendix)) OR (Appendix, Omental)) OR (Taenia Coli))) OR ((Liver) OR (Livers))) OR (Pancreas)) OR ((((Bile Ducts) OR (Bile Duct)) OR (Duct, Bile)) OR (Ducts, Bile))) OR ((Gallbladder) OR (Gallbladders))) OR ((((((Biliary Tract) OR (Tract, Biliary)) OR (Biliary System)) OR (System, Biliary)) OR (Biliary Tree)) OR (Tree, Biliary))) AND (specimen) AND (((((((((((((((Microbiota) OR (Microbiotas)) OR (Microbial Community)) OR (Community, Microbial)) OR (Microbial Communities)) OR (Microbial Community Composition)) OR (Community Composition, Microbial)) OR (Composition, Microbial Community)) OR (Microbial Community Compositions)) OR (Microbiome)) OR (Microbiomes)) OR (Human Microbiome)) OR (Human Microbiomes)) OR (Microbiome, Human)) OR (Microbial Community Structure)) OR (Community Structure, Microbial)) OR (Microbial Community Structures))) .

Through searching and screening, 977 articles were selected. Based on their relevance to the research topic, the credibility of the research methods, and the effectiveness of the research content, we gradually screened and excluded articles with poor research quality, reviews, and duplicate content. Finally, 92 articles were included (Fig. 1). Our team members carefully read and analyzed the content of the articles, and provided a systematic review and in-depth discussion of the research conclusions.

### Inclusion and exclusion criteria

In the literature review, select literature that meets the following criteria: (i) includes relevant studies on the sampling and detection of gut microbiota, and (ii) specifically refers to the oral cavity, esophagus, stomach, duodenum, jejunum, ileum, colon, liver, pancreas,

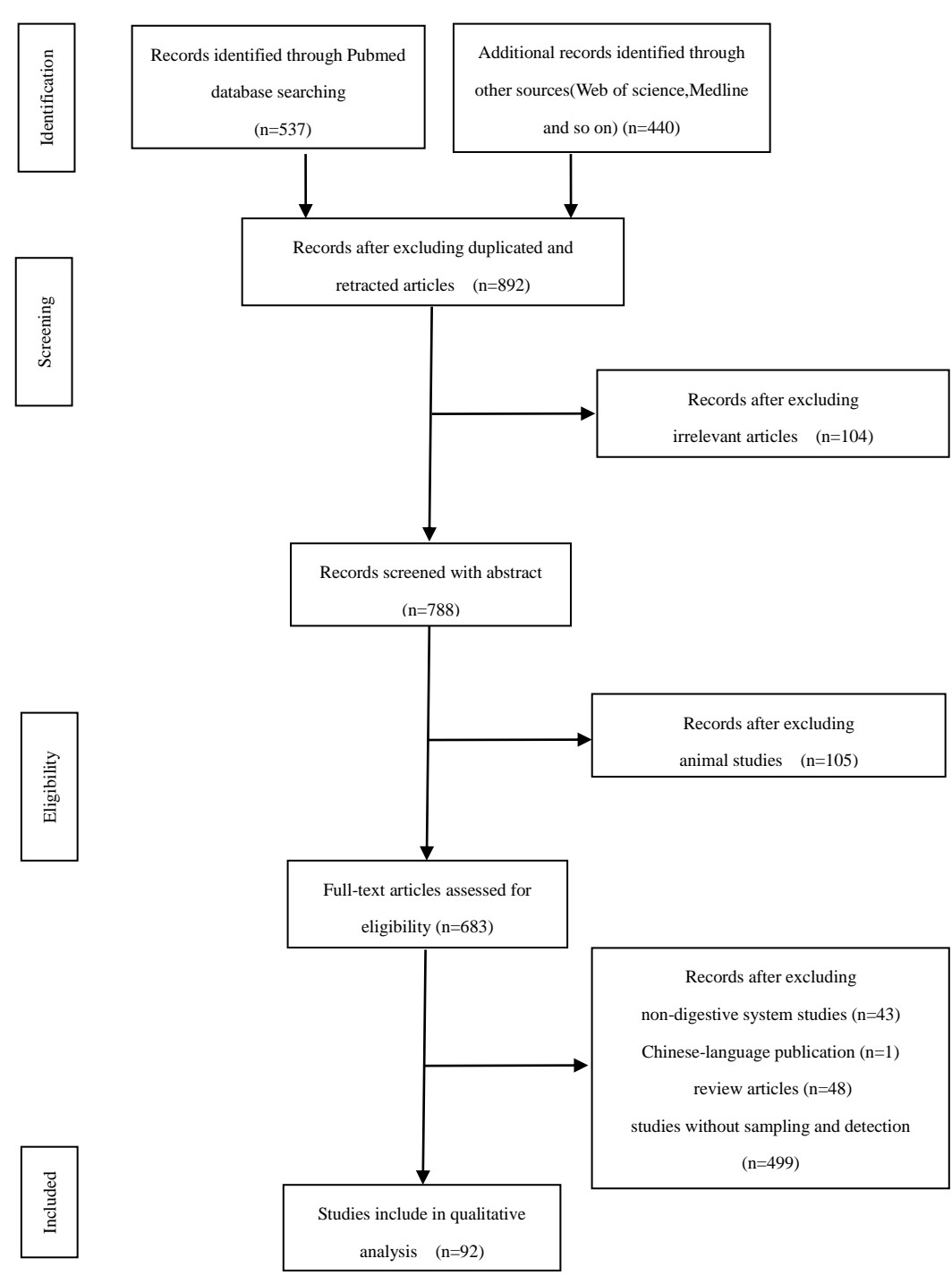

**Figure 1   Methods used in the study of digestive flora and their advantages and disadvantages.**

biliary, and bile duct; (iii) There are no strict limitations on the research methods for microbial communities. (iv) The language of literature was limited to English.

If the sampling method is not explicitly mentioned in the study, exclude the literature; Research on animals, human xenografts, and cancer cell lines have also been excluded. Exclude literature reviews, reviews, systematic reviews, or reader letters.

### Literature screening

Two reviewers (CX, LSZ) analyzed, determined, and scored all the selected literature according to their principles independently. Discussion or consultation to a third reviewer (WH) was conducted when difference in the decisions occurred.

### Data extraction and quality assessment

The information extracted from the literature by two authors (Liu and Gao), independently were: title, first author names, year of publication, country, and general situation of the included cases. The quality of each included observational study methodology was assessed using Newcastle-Ottawa Quality Assessment Scale (NOS) (*Li et al., 2016*). NOS included three aspects for cohort studies: selection, comparability, and exposure or outcome. A study having greater than or equal to six scores was considered as ahigh-quality study, while having nine stars is the full score (*Stang, 2010*; *Lo, Mertz & Loeb, 2014*).

## LITERATURE REVIEW

### Oral cavity

The oral cavity harbors the second-most diverse microbiota in the human body. This oral microbiota not only correlates with oral diseases but also exhibits connections with other gastrointestinal microbiota and malignant tumors (*Asili et al., 2023*; *Imai et al., 2022*). Notably, when compared with other anatomical regions, obtaining samples from the oral cavity is more straightforward. Various noninvasive methods, such as collecting saliva, oral mucosa samples, tissue biopsies, and gingival plaque, can be employed for oral microbiota research (Fig. 2). Accurate sample collection is fundamental for studying the relationship between oral microbiota and diseases because different collection methods can influence the accuracy of research findings. By optimizing sampling techniques, researchers can efficiently reveal the characteristics of oral microbial communities and their associations with pathological conditions.

Saliva is a valuable indicator of the overall composition of the oral microbiota. Saliva collected *via* the spit-out method contains five predominant bacterial phyla: *Bacteroidetes*, *Firmicutes*, *Proteobacteria*, *Fusobacteria*, and *Actinobacteria* (*Fan et al., 2018*). This method is simple and cost-effective, with a sampling process that minimally disrupts the oral environment. However, the procedure requires 1–5 min, and optimizing sampling duration may be critical for feasibility in large-scale cohort studies (*Gomar-Vercher et al., 2018*).

Oral rinse samples obtained through mouthwash share a microbial structure similar to that of saliva collected using the spit-out method, making them suitable for saliva collection (*Fan et al., 2018*; *Jo et al., 2019*). This approach is efficient and rapid, making it well suited for large-scale population sampling. It also proves beneficial for individuals

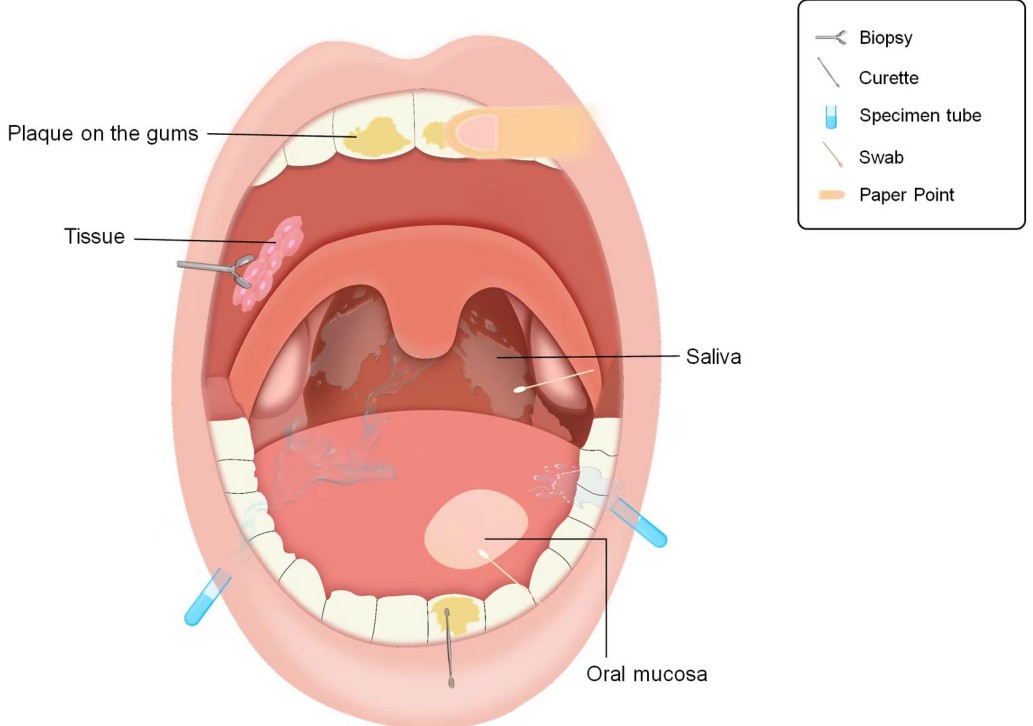

**Figure 2** **Approaches to obtaining microbial samples in oral microbiota research.** Sampling pattern diagram of oral microbiota.

such as the elderly, those with dry mouth, or those undergoing tumor radiotherapy and chemotherapy, who may struggle to produce saliva or expel it. It is important to note that Scope mouthwash contains ethanol, which may irritate the oral mucosa and could lead to acute alcohol poisoning if accidentally ingested by children.

A study conducted by *Omori et al. (2021)* from Osaka Medical College in Japan compared various saliva collection methods, including chewing, spitting, cotton swabbing, and rinsing. The results indicated that the microbial composition of saliva collected using the different methods exhibited significant similarity. In cases involving infants, young children, or individuals unable to rinse their mouths, saliva collected *via* cotton swabbing is sufficient for oral microbiota research. The research team suggested that differences in test results could be linked to whether saliva was rinsed before sampling and the chosen collection method; however, chewing or non-chewing had no discernible effect on the study of salivary flora. This conclusion aligns with the findings of another Japanese scholar *Jo et al. (2019)* and *Belstrometal (2016)* from Denmark. However, this contradicts other studies (*Gomar-Vercher et al., 2018*; *Zhu et al., 2020*), which assert that there are notable differences in the diversity and composition of the microbiota between chewed and non-chewed saliva obtained from the same individual. Additionally, the rate of salivary flow can influence the composition of the oral microbiota (*Lynge & Belstrom, 2019*), necessitating further investigation of salivary microbiota under varying stimulus conditions. Furthermore, the

choice of sampling kits and fixatives used in saliva sample storage may have influenced the results (*Vogtmann et al., 2019*; *Yano et al., 2020*).

Saliva samples are valuable for analyzing the overall characteristics and dynamic trends of oral microbiota, making them particularly suitable for large-scale population-based studies of oral microbial communities. When investigating the associations between oral diseases and systemic conditions, saliva serves as an effective preliminary screening specimen type, thereby providing a foundation for further in-depth investigations into these biological linkages. Considering the site-specific nature of microbiological composition within different regions of the oral cavity, it is important to recognize that saliva samples may not fully represent the local microbial community, particularly when dealing with localized lesions that require sampling from specific sites (*Yu et al., 2017*).

Tissue biopsies can offer a more precise representation of local mucosa-associated flora. However, this method is invasive and carries the risk of injury. Mucosal samples collected from oral swabs exhibit microbial community compositions similar to those of tissue biopsy samples, offering a practical and cost-effective alternative for biopsy purposes (*Hernández-Arriaga et al., 2019*). Notably, the paper disc method and curette method do not yield significant differences in detecting three "red complex" species in subgingival biofilm sampling of periodontitis patients. However, the paper disc method is more adept at detecting the presence of actinomycetes, which is crucial for determining whether antibiotic treatment is required (*Belibasakis, Schmidlin & Sahrmann, 2013*).

Dental plaque collection requires specialized tools, such as sterilized explorers or filter paper strips, and is typically performed by licensed dental professionals to obtain site-specific biofilm samples. Meanwhile, gingival crevicular fluid sampling employs periodontal probes or microcapillary tubes to aspirate inflammatory exudate from the gingival sulcus. Although technically demanding, this method yields specimens with higher histopathological relevance to periodontal tissues. Both collection modalities are essential for etiological investigations and clinical diagnostics of oral pathologies, including dental caries and periodontal diseases, providing targeted microbiological evidence to guide therapeutic interventions. These site-specific sampling approaches prove particularly valuable in localized ecological studies of the oral microbiome, enabling precise characterization of niche-specific microbial community variations and symbiotic/competitive interactions.

In summary, the choice of the oral sampling method should be tailored to the specific goals of the study and the clinical context in which it is conducted. Research on the oral microbiota should consistently employ standardized sampling methods, including uniform stimulation conditions, collection devices, and storage protocols, to minimize variation and ensure reliable results.

## Esophagus

The esophagus was previously believed to be devoid of microbial inhabitants, functioning solely as a transient conduit for bacteria traveling from the mouth to the stomach, or as a region influenced solely by gastric reflux. However, recent research has conclusively established the presence of distinct resident microbiota within the esophagus. Notably,

the microbiota in different segments of the normal esophagus exhibit similarities (*Yin et al., 2020*). The esophagus presents unique sampling challenges due to its rapid transit time, limited sample availability, and susceptibility to contamination from oral/gastric contents. Current esophageal sampling approaches comprise invasive and noninvasive methodologies. Invasive techniques include endoscopy-guided brush cytology and biopsy forceps sampling, whereas noninvasive alternatives encompass the Cytosponge cell collection device, cellular sponge sampling systems, and esophageal secretion retrieval methods. The selection of sampling protocols critically affects specimen integrity and yield, which in turn influences the reliability of downstream microbial analyses and molecular diagnostics. Characterization of the esophageal microbiota relies primarily on biopsy tissue and mucosal samples. Among the various sampling methods for esophageal microbiota, mucosal biopsy and mucosal brush or swab techniques are commonly employed (*Yin et al., 2020*), with mucosal biopsy considered the preferred modality (*Okereke et al., 2019*). However, it is important to acknowledge certain limitations associated with mucosal biopsy, including the potential to miss low-abundance or rare bacteria and the risk of complications such as bleeding and perforation during the procedure. These factors render it unsuitable for repeated or frequent samplings.

Mucosal swabs, on the other hand, can yield bacterial genera similar to those obtained *via* tissue biopsy. Research by *Gall et al. (2015)* suggested that esophageal biopsies contain substantial amounts of both human and bacterial DNA (100000:1). Mucosal swabs not only capture the full spectrum of bacteria present in biopsies but also enhance the recovery rate of bacterial DNA while reducing the proportion of human DNA. This leads to improvements in the quantity and quality of bacterial DNA. Studies conducted by *Li et al. (2020)* and *Liu et al. (2019a)*, *Liu et al. (2019b)* found that the microbial diversity in esophageal mucosal swab samples was higher than that in biopsy samples, with the latter observing a higher abundance of *Weilonella spp.* in ESCC paraneoplastic tissue swab samples than in biopsy samples. It is worth noting that while esophageal brush and swab sampling can minimize the risk of bacterial contamination from the oropharynx and stomach, they both rely on endoscopy, which demands proficiency in endoscopic techniques, is relatively costly, and is invasive.

The Cytosponge is a compressed mesh sponge attached to a string, encapsulated in ingestible gelatin measuring a mere two cm in length, like a large pill (*Iqbal et al., 2018*). When the gelatin dissolves, the sponge expands and can collect cells from the esophagogastric junction and throughout the esophagus when pulled out through the mouth using the string. The device can potentially detect a wide array of esophageal pathology for a cost significantly lower than those of endoscopy and biopsy (*Iqbal et al., 2018*). The Cytosponge device offers an alternative for esophageal microbiota sampling that circumvents the need for an endoscope and can be completed within 5–7 min. Moreover, it yields microbial DNA quantities over ten times higher than those obtained from biopsy tissue and endoscopic brushes. Severe atypical hyperplasia represents an early lesion of esophageal adenocarcinoma, and Cytosponge equipment can detect changes in the esophageal microbiota during this stage (*Elliott et al., 2017*).

The device consists of a weighted gelatin capsule containing 90 cm of nylon string. The capsule is swallowed no more than 4 h before the scheduled endoscopic procedure, and the proximal end of the string is taped to the subject's cheek. One hour after swallowing the capsule, the string is removed, the esophageal segment is harvested and placed in an esophageal string test (EST) elution buffer, and the eluate is frozen for EAP analyses, as described previously (*Furuta et al., 2013*; *Ackerman et al., 2019*). EST allows for the sampling of esophageal luminal secretions and sloughed epithelial and inflammatory cells (*Muir et al., 2022*). EST is another method for obtaining flora comparable to mucosal biopsy without relying on endoscopy or anesthesia, making it well suited for subjects, especially children, due to its tolerability (*Fillon et al., 2012*). Additionally, EST can provide longitudinal samples to monitor microbiota changes during the treatment of esophageal disorders (*Harris et al., 2015*). Both the Cytosponge device and EST offer noninvasive sampling, have extensive sampling surface areas, lower equipment costs, and can be repeated. However, it is important to note that they collected samples from the entire esophagus, which may introduce contamination from bacteria originating in the oral cavity and proximal stomach during sampling. Notably, a prospective study (*Jung et al., 2022*) from South Korea found that the number of bacteria (operational taxonomic units) in esophageal cavity fluid aspirated through endoscopy in patients with esophageal achalasia were significantly higher than that in esophageal biopsy tissue samples. It is important to recognize that both esophageal lavage and aspirate sample the bacterial community within the esophageal lumen and may not capture certain bacteria that adhere to the esophageal mucosa.

Invasive techniques yield high-quality esophageal microbiota specimens with superior diagnostic resolution but entail greater patient discomfort and procedural complexity. By contrast, noninvasive methodologies offer enhanced patient compliance and operational simplicity, though constrained by lower specimen yield and susceptibility to extraneous factor interference. Clinically, invasive approaches are prioritized for mechanistic investigations requiring precise microbial profiling, whereas noninvasive strategies prove advantageous for population-level epidemiological surveillance and preliminary screening. Methodological optimization is paramount because divergent sampling protocols may generate conflicting microbiota signatures, potentially confounding the interpretation of esophago-microbial pathophysiology in disease states.

## Stomach

The stomach is traditionally perceived as an inhospitable environment for microbial growth owing to its unique acidic conditions and other inherent antibacterial mechanisms. However, with the discovery of *Helicobacter pylori* (HP) and continuous advancement of microbial research, an increasing number of gastric endobacteria have been identified. Dysregulation of the gastric microbiota may play a key role in the entire oncogenic process, from precancerous lesions to gastric malignancies (*Zeng et al., 2024*). The composition of the gastric flora varies in different gastric disorders (*Stewart, Wu & Chen, 2020*; *Bessede & Megraud, 2022*).

Currently, biopsy samples, mucosal samples, and gastric juice are the primary sources used to study gastric microbiota. Although fecal samples have been used for HP testing and screening, they do not provide a comprehensive representation of the entire gastric microbiota. Endoscopic biopsy tissue is commonly employed to investigate gastric microbiota, as it can effectively capture relevant microbiota compared to gastric fluid (*Sung et al., 2016*). Research has demonstrated that microbial composition differs between various anatomical segments of the stomach in the same disease, as well as between different diseases (*Deng et al., 2021*). Furthermore, variations exist in microbial community structure within different gastric microbial environments in the same patient. For example, the microbial community structures in normal tissues, adjacent tissues, and cancer tissues of patients with gastric cancer may not be entirely consistent (*Liu et al., 2019a*; *Liu et al., 2019b*). Biopsies, which are typically taken locally, have a limited sampling area, and patients taking medications such as anticoagulants or antiplatelet agents may face challenges with sampling due to the risk of bleeding. Consequently, biopsy samples may not comprehensively represent the entire gastric microbiota, limiting their applicability in certain patient microbiota studies.

Mucosal specimens, on the other hand, involve relatively minimal trauma, and the sampling range when scraping gastric mucosal specimens with biopsy forceps is controllable. Even for individuals taking anticoagulants, sampling is safe, and mucosal specimens exhibit high sensitivity for diagnosing HP infections (*Matsumoto et al., 2016*). Endoscopic cytology brushes for mucosal specimen collection offer similar advantages and can replace biopsy forceps, albeit with the risk of contamination. Sheathed endoscopic cytology brushes can reduce cross-contamination during sampling (*Kashiwagi et al., 2020*; *Voss et al., 2022*). *Graham et al. (2005)* developed a transoral device with a protective sheath to brush gastric mucosal samples for HP testing, achieving a 100% bacterial recovery rate. They considered this device to be reliable, rapid, minimally invasive, independent of an endoscope, less costly, and suitable for outdoor field sampling in remote areas.

Although transient oral and esophageal bacteria may be present, gastric fluid is uniformly distributed throughout the stomach and can reflect the intragastric microecological environment (*Imai et al., 2022*). Gastric fluid specimens can be obtained using nasogastric tubes and endoscopic aspiration. However, specimens obtained by nasogastric tubes may contain food and have a higher risk of oropharyngeal contamination, whereas endoscopic aspiration reduces this risk. A study (*Sung et al., 2016*) conducted by Seoul National University in South Korea compared the microbial composition of the gastric antrum, gastric mucosal samples, and gastric juice samples. The results indicated differences in microbial composition between gastric juice aspirated through endoscopy and biopsy samples. Although gastric juice exhibited higher microbial diversity and operational taxonomic units, gastric mucosal samples had higher abundances of HP and Aspergillus, suggesting that gastric mucosal tissue samples can provide more meaningful insights into gastric microbiota. *John et al. (2022)* at the University of Nebraska–Lincoln in the United States developed an innovative sampling technique: a nanofiber capsule composed of nanofiber rectangles and spheres produced through electrostatic spinning, gas foaming, coating, and cross-linking. This device facilitates the collection of biological samples from

internal organs such as the jejunum, stomach, esophagus, and oropharynx, without the need for sedation. Because of its high adsorption capacity, this method provides a sufficient sample volume that is easily recoverable. Furthermore, it is minimally invasive, rapid, cost-effective, and suitable for population-based sampling.

The string test can collect gastric fluid samples for HP detection, and when combined with quantitative PCR, can also identify HP resistance to antibiotics. This method can be considered when a biopsy or gastric fluid aspiration is insufficient for sampling. String test examination has the advantages of convenience, economy, and good tolerance by the examinee. However, some disadvantages may limit the clinical application of the technique. For instance, the detection of HP is not as sensitive as that of endoscopy, the time-consuming test may interfere with the daily life of the examinee, and the test is difficult to execute in young children. Nevertheless, reducing the size of the capsule may improve the success rate of the test (*Han et al., 2023*; *Leong et al., 2003*; *Tafur et al., 2018*).

The study of gastric microbiota holds significant potential for developing novel diagnostic biomarkers and therapeutic approaches, thereby enhancing treatment efficacy for gastric disorders and long-term patient outcomes. While invasive techniques, such as endoscopic mucosal biopsy, yield high-quality specimens, these procedures present inherent limitations, including patient discomfort, operational complexity, and substantial healthcare costs. Conversely, noninvasive sampling methods offer procedural simplicity and minimal patient discomfort, yet they demonstrate reduced reliability in sample representativeness and diagnostic accuracy. Current methodological constraints underscore the imperative for future research to focus on advancing refined noninvasive sampling technologies. Such innovations should aim to achieve precise acquisition of gastric microbiota specimens while concurrently optimizing patient comfort and procedural safety parameters.

## Small intestine

The small intestine, comprising the duodenum, jejunum, and ileum, represents the longest segment of the gastrointestinal tract and harbors a distinct microbial community that diverges significantly from both fecal microbiota and microbial populations in other digestive regions. This unique microbial composition is associated with nutrition and various other diseases (*Kastl et al., 2020*; *Leite et al., 2020*; *Nagasue et al., 2022*). The microbiota in different parts of the small intestine plays different roles in nutrient absorption, immune regulation, and disease development and requires targeted sampling. The small intestinal microbiota is easily affected by upper gastric acid and lower colon reflux, making sample collection difficult and requiring optimization of sampling methods. Invasive sample collection methods include double balloon enteroscopy, single balloon enteroscopy, and capsule endoscopy-assisted sampling. Noninvasive collection methods comprise jejunal puncture drainage fluid sampling, small intestinal microbiota detection in fecal transplant donor screening, and small intestine fluid collection capsule technology.

Conventional gastroscopy is unable to comprehensively examine the small intestine, necessitating the use of endoscopy. However, endoscopy of the small intestine is time-consuming, expensive, and not widely available in some healthcare facilities. Capsule
endoscopy lacks the ability to collect and store samples, making it challenging to directly sample the small intestine. Consequently, the key to studying the small intestine microbiota lies in the development and optimization of small intestine sampling techniques.

As early as 2014, *Huse et al. (2014)* collected paired mucosal biopsies and mucosal brush samples from patients with ulcerative colitis who had undergone colon resection and ileal pouch anal anastomosis. The results revealed that the bacterial community structures of the two groups were similar; however, the mucosal brush samples had a higher yield of bacterial DNA, covered a wider sampling area, and caused less trauma. *Dreskin et al. (2021)* used esophagogastroduodenoscopy (EGD) suction catheters to extract duodenal fluid while simultaneously obtaining duodenal tissue samples with biopsy forceps for a comparative study of the microbial composition. The findings showed significant differences in the microbial composition between the duodenal suction fluid and biopsy samples, suggesting that a comprehensive analysis of the duodenal microbiota can be achieved by combining both approaches. *Leite et al. (2020)* from the Cedars Sinai Medical Center in Los Angeles reported differences in the bacterial community of duodenal aspirates obtained through EGD sterile suction catheters when compared to fecal samples. The team also utilized anterograde double-balloon endoscopy to collect intraluminal fluid from the duodenum, jejunum, and furthest segment of the small intestine accessible *via* EGD for microbial composition analysis. The results indicated variations in microbial community characteristics among these segments. Compared with standard biopsy forceps and sterile brushes Brisbane aseptic biopsy device (BABD) sampling demonstrated the ability to minimize cross-contamination during sampling (*Shanahan et al., 2016*). Balloon-assisted enteroscopy (BAE), a novel endoscopic technique, enables sampling of the entire small intestine and is suitable for studying the small intestinal microbiota. *Nagasue et al. (2022)* used anterograde BAE endoscopic brushes to collect mucosal samples from various segments, and retrograde BAE endoscopic brushes to sample the proximal ileum, distal ileum, and rectum. These findings underscore the distinct nature of the small intestine microbiota compared to that of feces and other parts of the digestive tract. BAE relies on endoscopic examination and is invasive, with high examination costs. The depth of small intestine access depends on the subject's physical condition and endoscopist's expertise, and intestinal preparation is necessary before retrograde BAE examination, which, to some extent, limits its application.

*Waimin et al. (2020)* designed a novel 3D sampling capsule that does not require batteries. This device could enter the small intestine to collect bacterial samples for subsequent cultivation. It offers low-cost, non-toxic, harmless sampling with high patient comfort, and does not require a clinical environment. However, this is currently in the development and improvement stage. Duodenal capsules are safe, straightforward, and suitable for sampling the duodenal microbiota in children (*Gracey, Suharjono & Sunoto, 1977*). Collecting mucosal samples during small intestine surgery can accurately reflect the relationship between microbial communities and the spatial structure of the digestive tract, while avoiding contamination from other regions. However, this method is unsuitable for healthy individuals (*Villmones et al., 2022*). Effluent from a stoma after enterostomy has also been used for small intestinal microbiota analysis, but there is a risk of contamination

from the adjacent skin and the external environment (*Zoetendal et al., 2012*). Given the varying capabilities of different devices for small intestine examinations and the diverse locations of lesions, it is essential to select appropriate sampling methods based on specific research requirements and patient conditions. For improving the accuracy and reliability of sampling, high-precision sampling techniques need to be developed in the future to obtain accurate samples of small intestinal microbiota, thereby providing strong support for exploring the composition and function of small intestinal microbiota.

## Colorectum

The colorectum harbors the richest microbiota within the digestive tract and its composition varies across different regions (*Kwon et al., 2021*). Various samples are commonly used for colorectal microbiota research, including feces, biopsy tissue, mucosal samples from the cavity wall, and intestinal lavage fluid. Sampling methods include direct fecal collection, rectal swabs, endoscopic biopsies, mucosal brushing, and aspiration of intestinal luminal lavage. Colorectal sampling is generally more straightforward than small intestine sampling. However, owing to the presence of physiological curvatures, there is a high demand for advanced endoscopic sampling techniques.

Fecal specimens are easily obtainable and have been extensively employed in studies of gut microbiota. A cohort study in Iran demonstrated that the microbial community composition of fecal samples collected using fecal collection kits and fecal occult blood test (FOBT) cards were similar and remained stable for up to four days at room temperature. Additionally, using FOBT cards for fecal sample collection in microbial community research has proven to be cost-effective (*Wu et al., 2021*). Fecal swab samples maintain detection sensitivity at 4 °C and can be a suitable replacement for feces in intestinal microbiota research. They received US Food and Drug Administration (FDA) approval for intestinal bacterial culture (*Richard-Greenblatt et al., 2020*). However, it should be noted that some individuals are reluctant to handle feces, and fecal collection may not always be feasible, especially for elderly individuals with constipation and critically ill patients.

Studies have shown that the gut microbiota composition in fecal samples from healthy individuals closely resembles that of the corresponding rectal swabs, with comparable performance in metabolomics and gut function studies (*Radhakrishnan et al., 2023*; *Reyman et al., 2019*; *Turner et al., 2022*). Bacterial culture of rectal swab samples also offer a significantly shorter detection time than loose feces, which can reduce patient hospitalization time (*Jean et al., 2019*). Rectal swab sampling is feasible for patients to collect at home or by medical staff in hospitals, and is suitable for clinical disease diagnosis and large-scale research (*Budding et al., 2014*). For infants, critically ill patients, and those unable to provide fecal samples, rectal swabs can be collected without the need for intestinal preparation and can serve as an alternative method for fecal testing. Nevertheless, rectal swab samples are susceptible to contamination from perianal skin and may be influenced by sampling time, which requires careful consideration when using this method (*Chanderraj et al., 2022*; *Fair et al., 2019*; *Schlebusch et al., 2022*). Some researchers have suggested that glove tip samples can also be employed to study gut microbiota, potentially replacing rectal swabs and fecal sampling (*Short et al., 2021*). However, regardless of whether fecal
samples are collected directly using the swab method or the fingertip method, they cannot fully represent the local lumen and mucosal surface microbiota composition at different locations of the colon and rectum. Factors such as gastric acid, bile acids, oxygen levels, and antimicrobial peptides may influence the alpha diversity of the gut microbiota across the upper and lower gastrointestinal tract (*Simren et al., 2013*). *Zhang et al. (2014)* demonstrated unique spatial heterogeneity in mucosal-associated microbial communities and species along the intestinal tract. A study by *Zilberstein et al. (2007)* on healthy individuals revealed distinct regional variations: *Fusobacterium* abundance was higher in the rectum, *Streptococcus* dominated the sigmoid colon, *Enterococcus* was enriched in the transverse colon, and *Bacteroides* predominated in the proximal colon but declined in the sigmoid colon and rectum. By contrast, *Jiao et al. (2022)* reported no significant differences in richness, evenness, community composition, or taxonomic structure across colonic segments, with *Firmicutes* (47%), *Bacteroidetes* (39%), and *Proteobacteria* (6%) being the dominant phyla, followed by *Verrucomicrobia*, *Fusobacteria*, *Desulfobacterota*, and *Actinobacteria*. Notably, significant differences in biodiversity and taxonomic structure were observed between rectal and fecal bacterial communities.

Colonoscopy can be utilized for research on mucosal microbiota in various intestinal biopsies. However, it has limitations related to its invasiveness when obtaining a large number of samples simultaneously. Colon lavage enables the collection of lavage fluid from different intestinal segments, which can accurately represent the microbial composition of colon biopsy specimens. This provides a larger sample size and can serve as an alternative method when the biopsy sample size is limited (*Kwon et al., 2021*; *Watt et al., 2016*). A study (*Araújo-Pérez et al., 2012*) conducted at the University of North Carolina Hospital found that the diversity of bacterial communities in rectal swab samples from subjects who did not undergo intestinal preparation was higher than that in rectal biopsy tissue. The microbial community structure may exhibit short-term changes before and after intestinal preparation, which should be considered (*Shobar et al., 2016*; *Zou et al., 2023*). *Kwon et al. (2021)* reported that rectal swab samples collected after intestinal preparation are similar to the mucosal microbiota represented by colonic lavage fluid, but differ in microbial composition from fecal samples collected before intestinal preparation. *Matsumoto et al. (2019)* suggested that mucosal brush samples could aid in the analysis of mucosal-associated microbiota, which differs from fecal samples. Although endoscopic biopsy, colonic lavage, mucosal brushing, and rectal swabs can all yield mucosal-associated microbiota samples, each sampling method has its own set of advantages and disadvantages. Colonic lavage and mucosal brushing is non-invasive, provides control over the sampling area, and yields more biological samples than biopsy. However, all sampling methods, except for rectal swabs, rely on endoscopy and are invasive, whereas rectal swabs are susceptible to contamination by intestinal contents, necessitating caution during collection. The sampling schematic of gastrointestinal microbiota is shown in Fig. 3.

Invasive methods may cause trauma to patients, are complex to operate, carry high risks, require professional equipment and operators, and have poor patient tolerance. On the contrary, noninvasive methods are noninvasive or minimally invasive, with good patient tolerance and low risk, but may not be able to obtain high-quality samples. Therefore, when

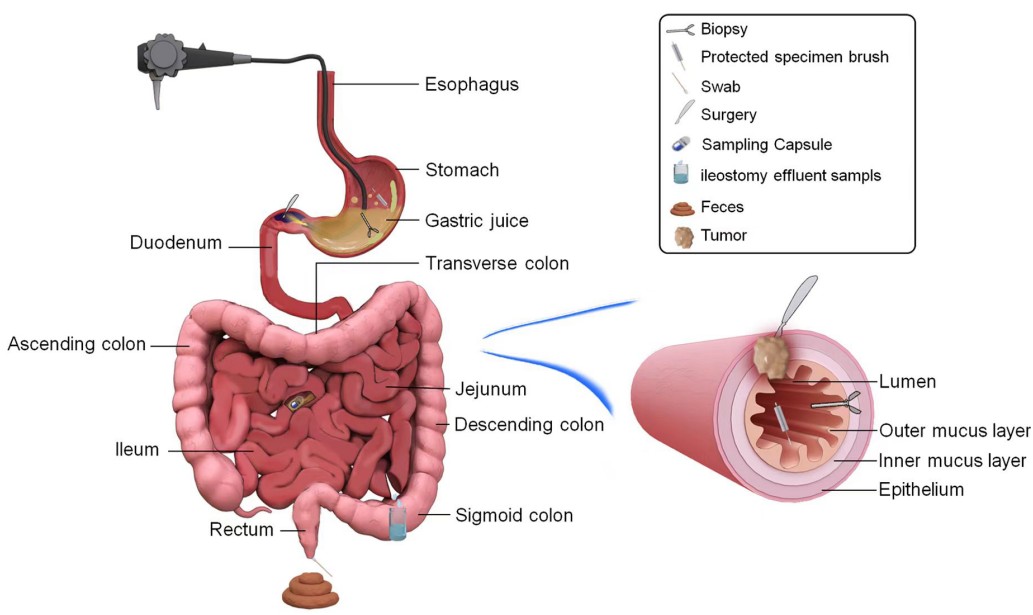

**Figure 3 Pathways to obtain microbial samples in the study of biliary and pancreatic microbiota.** Pattern diagram of pancreatic and hepatobiliary microbiota.

sampling, the conditions of the sampling structure and the patient's physical condition must be considered comprehensively, and an appropriate sampling method should be chosen. In the future, more noninvasive and high-precision sampling techniques need to be developed to reduce patient pain and injury. In addition, the accuracy of sample analysis techniques needs to be improved to more comprehensively and accurately analyze the composition and function of colorectal microbiota.

## Biliary tract, liver, and pancreas

The bile duct, liver, and pancreas are considered the extraluminal organs of the digestive tract, and their disease status is closely associated with local flora dysbiosis. Common sample types for research on the microbiota of the liver, gallbladder, and pancreas include surgically resected tissue, tissue biopsy, and bile samples (Fig. 4). These samples can be obtained through surgical procedures, endoscopic retrograde cholangiopancreatography (ERCP), percutaneous transhepatic cholangiography drainage (PTCD), endoscopic ultrasound fine-needle biopsy (EUS-FNB), and other methods.

## Biliary tract

The biliary tract includes the gallbladder and bile ducts, which are connected to the duodenum by the sphincter of Oddi. The Oddi sphincter plays a role in preventing the invasion of intestinal bacteria, and the bile itself exhibits antibacterial properties. The continual flushing of bile and the antibacterial effect of bile salts together contribute to the inhibition of bacterial growth in the biliary system. Consequently, the biliary system of healthy individuals has historically been regarded sterile. The biliary system is located deep and difficult to reach through conventional means, which poses great difficulties for sample

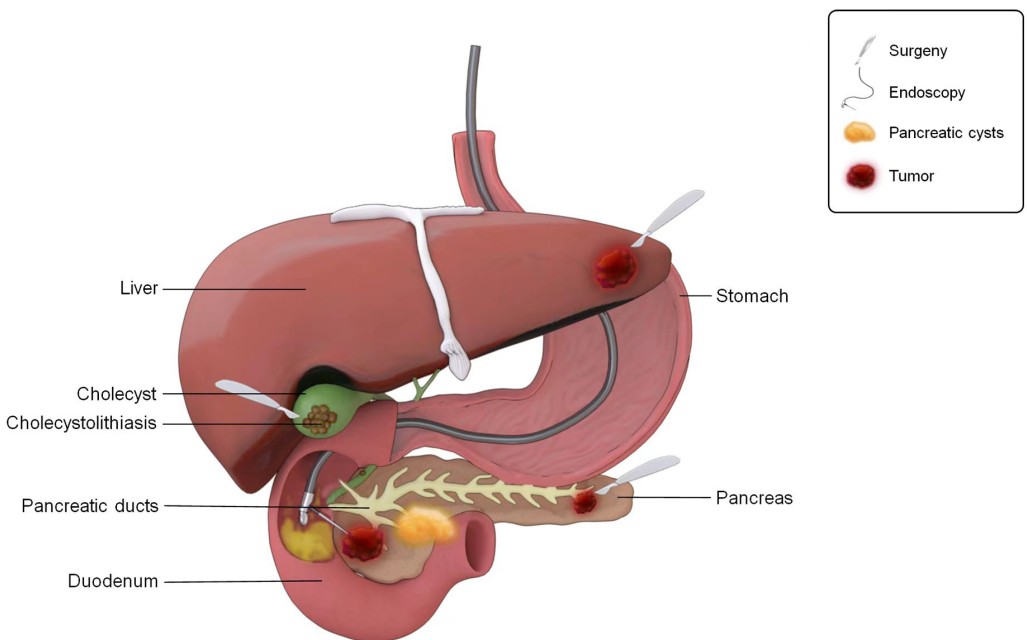

**Figure 4   The ways to obtain microbial samples in the study of gastrointestinal microbiota.** Sampling pattern diagram of gastrointestinal microbiota.

collection. Thus, the sampling method for the biliary tract must have good accessibility to ensure effective sample acquisition.

In 2014, Spanish scholars were the first to detect live bacteria and bacterial proteins in gallbladder bile, gallbladder mucus, and gallbladder biopsy tissues from healthy hosts (sows) (*Jiménez et al., 2014*). Subsequently, *Molinero et al. (2019)* reported the composition of the bile microbiota in liver transplant donors without liver and gallbladder diseases, with the primary phyla being *Firmicutes, Bacteroidetes, Actinobacteria*, and *Proteobacteria*. Differences were observed in the microbiota and bacterial metabolism profiles compared with those in patients with cholelithiasis. Notably, because liver grafts may undergo brief antibiotic treatment, their bile microbiota may not be fully representative of that of healthy individuals.

More recently, *Gookin et al. (2023)* did not identify any gallbladder core microbiota in gallbladder bile extracted from healthy dogs. The aforementioned research samples were obtained through surgery or euthanasia of the animals. Although surgical resection to obtain samples is suitable for patients with concurrent biliary diseases, it is not ethical or appropriate for healthy individuals. Thus, there are current challenges in studying gallbladder microbiota in normal individuals owing to ethical constraints in sample acquisition.

Bile samples for microbiota studies can be obtained through endoscopic access during RCP procedures. *Kim et al. (2021)* utilized ERCP combined with endoscopic nasobiliary drainage (ENBD) to extract bile microbiota from patients with brown pigment stones. The results revealed that bile contains a rich microbial community, with the highest abundance

of *Proteobacteria* at the phylum level, reaching 61.7%, which is similar to the composition of duodenal microbiota. However, it's important to note that ERCP procedures may disrupt the normal anatomy and functional barrier of the biliary tract, increasing the risk of infection in the ascending biliary tract. The *Effenberger et al. (2023)* team believed that bile sampling under endoscopic retrograde cholangiography (ERC) is feasible, as there may be residual bacteria in the oral cavity and endoscopy equipment that could be transferred to the biliary tract during the examination, potentially leading to contamination of the biliary system. However, this contamination does not appear to significantly affect the clinical results of the participants. Bile extraction *via* percutaneous transhepatic cholangiography drainage can also be used for microbial community research (*Gu et al., 2020*). *Gookin et al. (2023)* emphasized the risk of contamination in both gallbladder and ERCP bile extraction from liver transplant donors. Disinfecting the skin and extracting bile through ultrasound-guided percutaneous puncture or using sterile gloves during open surgery is recommended to minimize contamination.

For patients requiring definitive assessment of biliary microbiota and disease associations, particularly in cases of cholangitis or cholangiocarcinoma, invasive methods such as ERCP-guided bile aspiration remain the preferred approach. For those unable to tolerate invasive procedures, noninvasive methods like duodenal fluid sampling can serve as viable alternatives for initial screening and longitudinal monitoring.

## Liver and pancreas

Emerging evidence suggests that microbial communities may critically influence the pathogenesis and progression of hepatic and pancreatic diseases, necessitating comprehensive investigation into their functional roles and mechanistic pathways. The hepatic microbiota likely originates through dual anatomical routes—via portal venous circulation and hepatic arterial supply—presenting unique research challenges given its complex multisource derivation. Furthermore, the pancreas's status as a retroperitoneal organ with intricate ductal architecture, characterized by numerous branching conduits and tortuous pathways, creates substantial technical barriers in obtaining representative intraluminal microbial specimens.

Invasive methods for liver sample collection include sampling during liver transplantation surgery, percutaneous liver biopsy, and laparoscopic biopsy, whereas noninvasive methods consist of blood, fecal, and bile sample collection. The methods for collecting pancreatic samples include EUS-FNA, surgical resection specimen sampling, percutaneous pancreatic puncture, and invasive techniques such as ERCP to drain pancreatic juice. In addition, blood and feces can be used for pancreas-related research. For patients who need to clarify the relationship between pancreatic microbiota and diseases, invasive methods can be preferred to provide an important basis for disease diagnosis and treatment planning. For patients who cannot tolerate or do not require invasive procedures, noninvasive methods can be used for initial screening and monitoring.

Tumor tissues obtained through surgical resection or EUS-FNB can be utilized for studies that focus on tumor microbiota.The liver and pancreas have been found to harbor bacteria in various tumors, such as pancreatic and hepatocellular carcinomas

(*Nejman et al., 2020*). The composition of the flora within these tumors varies depending on the tumor type.

Chinese researchers have employed 16S rRNA sequencing to investigate the intratumoral flora structure in tissues from patients with intrahepatic cholangiocellular carcinoma who underwent radical resection (*Chai et al., 2023*). *He et al. (2023)* observed that the microbial diversity in hepatocellular carcinoma tissue excised during surgery was significantly higher than that in adjacent cancer tissues. The abundance of *Enterobacteriaceae, Fusobacterium,* and *Neisseria* increased significantly in hepatocellular carcinoma tissue, while the abundance of *Pseudomonas* decreased. Patients with different clinical characteristics exhibited significant differences in microbial and functional diversity within the tumor tissue.

In 2015, Japanese scholar *Mitsuhashi et al. (2015)* discovered that *Fusobacterium* status in formalin-fixed paraffin-embedded (FFPE) tumor tissue after pancreatic ductal adenocarcinoma (PDCA) was independently related to the prognosis of pancreatic cancer. The microorganisms found in resected pancreatic cancer tissues differed from those found in the whole pancreatic tissue of deceased individuals without pancreatic disease. Moreover, the flora in different parts of the pancreas (pancreatic head, tail, and duct) of the same healthy individuals resembled those of the duodenum (*Del Castillo et al., 2019*). It is worth noting that the tumor microbiota samples in the aforementioned studies were all obtained from patients with resectable tumors. Unfortunately, many patients miss the opportunity for surgery because of the advanced stage of the disease at initial diagnosis. Therefore, the findings of these tumor microbiological studies are limited, and the results may not represent the intratumoral microbiota of all patients with hepatic and pancreatic tumors.

*Masi et al. (2021)* found no significant differences in microbial diversity and taxonomic characteristics between surgical resection and EUS-FNB samples obtained from formalin-fixed paraffin-embedded samples of PDCA. This suggests that EUS-FNB can be effectively used in pancreatic microbiota research. This perspective has gained recognition among scholars (*Chu et al., 2022*; *Nakano et al., 2022*). Currently, it is believed that EUS-FNB can collect sufficient fresh cancer tissue without complications, making it an effective method for studying intratumoral microbiota in patients with PDCA in addition to surgical samples.

Bacteria have also been detected in pancreatic cyst fluids. Pancreatic cyst fluid obtained through EUS-FNA contains specific microbial groups (*Li et al., 2017*). Furthermore, undetermined bacteria exist in pancreatic fluid collected by a catheter from the pancreatic stump after pancreatic cancer surgery. This suggests that there may be an unexplored microbial world within the human body (*Okuda et al., 2022*). Currently, liver, gallbladder, and pancreatic sampling methods are invasive and carry the risk of contamination. In the future, it will be necessary to optimize these sampling methods. In addition to enhancing quality control, bioinformatics methods such as Deconam should be employed to eliminate potential DNA contamination (*Masi et al., 2021*). Decontam (https://github.com/benjjneb/decontam) is an open source R software package that improves the quality of macrogenomic and marker gene sequencing by identifying and removing contaminating DNA sequences (*Gentleman et al., 2004*). In addition, Decontam can be easily integrated with existing metagenomic sequencing (MGS) workflows, so researchers

can obtain more accurate microbial community profiles at little or no additional cost (*Davis et al., 2018*).

## CONCLUSIONS AND OUTLOOK

Various sampling methods offer the opportunity to collect samples from nearly all segments of the digestive system for studies of gastrointestinal tract microecology. The diversity of options caters to different populations and research objectives. When selecting specimens and sampling methods for microbiota studies, it is essential to consider the characteristics of the microbiota population, research environment, and the objectives of the study.

Saliva collection *via* expectoration and mouth rinse methods demonstrate comparable oral microbiota profiles. Expectoration is simple and cost-effective, while Scope mouthwash preserves bacterial genomic integrity, making it suitable for large-scale population sampling—particularly beneficial for individuals with xerostomia or difficulty producing saliva. Paraffin gum chewing effectively facilitates saliva collection for population screening but may alter salivary microbiota composition and is unsuitable for non-chewing populations. Oral swabs serve as a practical alternative for infants and individuals unable to rinse. Tissue biopsies, though invasive, enable targeted sampling of mucosa-adherent bacteria interacting closely with host tissues.

Esophageal tissue biopsies remain the gold standard for esophageal microbiota research but are invasive and unsuitable for repeated sampling. Sheathed endoscopy brushes and swabs yield microbiota profiles comparable to biopsies while being non-invasive, though both require endoscopic access. The esophageal string test offers low-cost, low-risk sampling with good patient tolerance, but may induce nausea/headaches and requires overnight hospitalization.

Biopsies remain critical for gastric/duodenal microbiota studies. Gastric fluid—distributed homogeneously throughout the stomach—provides a representative microenvironment profile. The string test demonstrates minimal invasiveness and excellent tolerability, particularly valuable for pediatric gastric sampling. Duodenal fluid collection remains experimental, while jejunostomy drainage carries contamination risks from adjacent skin and external environments.

Fecal samples are the standard non-invasive option for colorectal microbiota analysis. Compared to stool, endoscopic brush samples show higher Actinobacteria and lower Bacteroidetes abundance at the phylum level. Colonic lavage exhibits microbial diversity comparable to biopsies but yields significantly higher DNA quantities and microbial counts. However, both lavage and biopsy require invasive endoscopy, with lavage potentially capturing mixed luminal and mucosal communities.

Bile samples obtained *via* ERCP or cholecystectomy typically reveal dominant phyla: Firmicutes, Bacteroidetes, Actinobacteria, and Proteobacteria. Liver tissue—often surgically acquired—demonstrates higher microbial richness in

tumor *versus* adjacent tissues. Pancreatic sampling remains challenging, limited to surgical specimens or EUS-guided FNA/FNB, none being feasible for healthy controls.

The current sampling methods have limitations, including invasiveness, susceptibility to contamination, and segmental sampling. We systematically summarize the microbial characteristics, strengths, and limitations of these sampling approaches in Table 1.

Uniform sampling protocols should be followed to ensure consistency in microbiota studies. These protocols should include standardized stimulation conditions, storage equipment, temperatures, and analytical methods to minimize variability. Standardization of sample processing and preservation should be strengthened. First, unified sampling standards, including sampling methods, sampling locations, sampling times, etc., should be established to ensure comparability between different studies. Different research institutions should follow the same sampling standards to improve the consistency of research results. Second, the training of sampling personnel should be strengthened. They should master the correct sampling methods and operating procedures to ensure the accuracy and reliability of sampling. Third, the storage conditions and time of the samples need to be standardized, including factors such as temperature, humidity, and light, to ensure their quality and stability. Each research institution should follow the same sample preservation standards to ensure the integrity and availability of the samples.

Therefore, sample processing techniques should be improved, and strict quality control should be implemented during sampling to enhance sample quality and purity, reduce sample loss, and prevent contamination. By optimizing the sample preprocessing process, we can improve processing efficiency and quality. Precise microbiological analysis techniques should be chosen, and bioinformatics tools such as Deconam should be utilized to eliminate contaminating DNA sequences. New bioinformatic analysis tools can be developed in the later stage to better process and analyze large amounts of sequencing data, providing strong support for the study of gut microbiota. At the same time, multiomics analysis techniques can be combined to conduct in-depth research on the interaction between gut microbiota and hosts. Because the gut microbiota is subject to dynamic changes influenced by various factors, the effectiveness of different sampling methods in longitudinal flora studies needs to be assessed.

In the future, there is a need for further optimization and development of sampling devices to obtain more accurate microbiological samples, ensuring the reliability of the study results. Ideally, these sampling devices should allow for sampling from any part of the digestive tract without the risk of cross-contamination. They should also be convenient, cost-effective, minimally invasive, well tolerated by subjects, and suitable for large-sample population sampling. Additionally, ethical considerations related to the sampling process are of concern to researchers and should be appropriately addressed.

**Table 1  Methods used in the study of digestive flora and their advantages and disadvantages.**

| Organ | Sample source | Sampling | Main sequencing results | Disadvantage | Advantage | NOS |
|---|---|---|---|---|---|---|
| | | Spitting method or passive drooling (unstimulated saliva) | Main bacterial phyla: *Bacteroidetes, Firmicutes, Proteobacteria, Fusobacteria, Actinobacteria* (*Fan et al., 2018*). The 10 most abundant genera: *Streptococcus, Prevotella, Porphyromonas, Neisseria, Veillonella, Granulicatella, Actinomyces, Haemophilus, Rothia, Fusobacterium* (*Omori et al., 2021*). *Streptococcus* genus is most abundant in unstimulated saliva samples (*Gomar Vercher et al., 2018*). | Collection usually takes 1–5 min, and large-scale cohort studies require a shorter collection time. | Simple and cost-effective, the sampling process does not interfere with the oral environment, and is more favored in the field of proteomics than stimulating saliva samples. | 7, 8, 7 |
| | | Gargle | Oral bacterial spectrum similar to saliva samples obtained by spitting method (*Fan et al., 2018*). The microorganisms in the saliva collected by Scope mouthwash are usually stable. After being stored at room temperature for 4 days, the abundance of *Firmicutes* increases, while the abundance of *Proteobacteria* decreases (*Wu et al., 2021*). | Scope mouthwash contains ethanol, which may irritate the oral mucosa or cause acute alcoholism due to accidental ingestion by children. | Scope mouthwash has a broad-spectrum antibacterial effect, which quickly stops bacterial growth and preserves bacterial genomic information. Suitable for large-scale population sampling, not time-consuming, and for patients who cannot produce enough saliva, such as those with xerostomia or cancer patients undergoing radiotherapy and chemotherapy. | 7, 8 |
| | | Chewing (chewing paraffin glue) | At the genus level, *Streptococcus* occupies 20–35% of the total sequences in stimulated saliva, followed by *Neisseria* (7–25%), *Prevotella* (2–25%) and *Veillonella* (6–22%), *Fusobacterium* (<10%) and *Porphyromonas* (7%). *Fusobacterium* and *Porphyromonas* were two typical inhabitants of dental plaque. *Streptococcus* is the most abundant in the unstimulated saliva samples, many other bacterial genera are either at low proportion or absent when compared with stimulated saliva (*Gomar Vercher et al., 2018*). | The protein components in saliva samples will be diluted by stimulated saliva, and interference in the oral cavity may cause disturbance to the microbial structure of saliva. It is not suitable for subjects who cannot chew. | Convenient for rapid collection of saliva, comfortable, conducive to crowd screening. | 7 |
| | | Olfactory stimulation | Similar composition of salivary microbiota between olfactory stimulation and passive drooling (*Zhu et al., 2020*). | Low pH value, repeated collection can lead to significant changes in salivary peptide groups. | The sampling process has no direct interference with the oral cavity and is a suitable substitute for unstimulated saliva for peptide group and microbiota-related studies, facilitating rapid collection of saliva and population screening. | 8 |

**Table 1** (*continued*)

| Organ | Sample source | Sampling | Main sequencing results | Disadvantage | Advantage | NOS |
|---|---|---|---|---|---|---|
| | Saliva | Gustatory stimulation | Compared to unstimulated saliva, taste stimulation can lead to significant changes in the microbiota (*Zhu et al., 2020*). | The most evident drawback was the lower pH value. The flow rate is not constant, and interference in the oral cavity may cause disturbance to the microbial structure of saliva. | Convenient for rapid collection of saliva, comfortable, conducive to crowd screening. | 8 |
| | | Oral swab | The cotton swab method and the spitting method gave significantly different results. The oral swab method tends to contain less *Prevotella* and more *Haemophilus* at the genus level (*Omori et al., 2021*). | The sampling process has interference with the oral cavity. Centrifuge was required for post-collection processing. | Infants or patients who cannot produce saliva or have difficulty spitting are preferred. | 8 |
| | Tissue | Biopsy | Phyla pattern was similar between swabs and tissues. Main phyla: *Firmicutes*, *Proteobacteria*, *Bacteroidetes*, *Actinobacteria*. In 2 month old mice, *Cutibacterium acnes* was detected in higher abundance in the tissue biopsies (*Hernández Arriaga et al., 2019*). | Tissue damage, Higher cost | Can better screen for bacteria that adhere to the oral mucosal area and are in close contact with the host. | NA |
| | Oral mucosa | Oral swab | Phyla pattern was similar between swabs and tissues. Main phyla: *Firmicutes, Proteobacteria, Bacteroidetes, Actinobacteria*. In 15 month old animals, *Corynebacterium* was more abundant in the swabs. In 2 month old mice, *Streptococcus* danieliae was more abundant in swab (*Hernández Arriaga et al., 2019*). | The tested person may experience symptoms such as coughing, nausea, and even vomiting. | Cheap, practical, without the need for tissue cutting and the use of sedatives or sacrificial animals. It could collect bacteria present in higher abundances in the saliva, tongue, and shedding tissue surfaces. | NA |
| Oral cavity | Gingival plaque | Curette | The average bacterial counts and the numbers of *A.actinomycetemcomitans* obtained by the paper point or the curette sampling method did not show significant difference. *Aggregatibacter actinomycetemcomitans* 56% (36% positive and 20% negative), *Porphyromonas gingivalis* 96% (88% positive and 8% negative), *Treponema denticola* 98% (94% positive and 4% negative), *Tannerella forsythia* 96% (96% positive and 0% negative). Frequency of detecting targeted taxa using curette: *Aggregatibacter actinomycetemcomitans* 14%, *Porphyromonas gingivalis* 4%, *Tannerella forsythia* 2%, *Treponema denticola* 0%. (*Belibasakis, Schmidlin & Sahrmann, 2013*). | For local sampling, it cannot reflect the whole oral Microbiome. | It is more likely to achieve an efficient collection of bacteria from the tooth-adherent biofilm and an overall representation of the pocket microbiota. | 8 |
| | | Paper point | The Frequency of detecting targeted taxa using paper points: *Aggregatibacter actinomycetemcomitans* 30%, *Treponema denticola* 2%, *Tannerella forsythia* 2%, *Porphyromonas gingivalis* 0%. Compared with the curette, the paper point detected significantly higher levels of the three "red complexes". Paper point can more consistently detect the presence of A. *Actinomycetemcomitans* in patients with invasive periodontitis than the curette method. (*Belibasakis, Schmidlin & Sahrmann, 2013*) | For local sampling, it cannot reflect the whole oral Microbiome. | Paper point is more likely to provide a better representation of the outer biofilm layer or the 'free-floating' bacteria in the pocket. Higher levels were consistently detected in samples collected by paper point than by curette sampling. It is crucial to decide whether to use antibiotics as an adjunctive treatment for periodontitis. | 8 |

**Table 1** (*continued*)

| Organ | Sample source | Sampling | Main sequencing results | Disadvantage | Advantage | NOS |
|---|---|---|---|---|---|---|
| | Tissue | Biopsy | (Normal, Esophagitis, LGIN, HGIN, ESCC) The top 5 bacterial phyla: *Firmicutes*, *Proteobacteria*, *Bacteroidetes*, *Actinobacteria*, *Fusobacteria*. The top 10 genera: *Streptococcus*, *Prevotella*, *Veillonella*, *Actinobacillus*, *Haemophilus*, *Neisseria*, *Alloprevotella*, *Rothia*, *Gemella*, *Porphyromonas* (*Li et al., 2020*). | | | 9 |
| | | | (The carcinoma and adjacent normal tissue) phylum level:*Bacteroidetes*, *Firmicutes*, *Proteobacteria*, *Fusobacteri*, *Actinobacteria*; top ten genera: *Prevotella*, *Streptococcus*, *Veillonella*, [*Prevotella*], *Haemophilus*, *Capnocytophaga*, *Fusobacterium*, *Selenomonas*, *Peptostreptococcus*, *Neisseria* (*Liu et al., 2019a*; *Liu et al., 2019b*). | Invasive, with a risk of bleeding and perforation, limited sampling surface area, unsuitable for repeated and frequent sampling, requiring anesthesia | Still the gold standard for analyzing the microbial community in the esophagus. | 8 |
| | Mucosa | Endoscopic cytology brush | The endoscopic brush consistently has all of the OTUs found in the biopsy samples, as well as additional OTUs. In the Barrett esophagus cohort, *Streptococcus* and *Prevotella* are dominant in the upper gastrointestinal tract (*Gall et al., 2015*) | Endoscopy dependent, with a very high ratio of human to bacterial DNA | enriching the abundance and diversity of bacteria, improving the quantity and quality of recovered bacterial DNA. | 8 |
| | | Swab | The swab mucosa and tissue biopsy specimens have similar microbial profile (*Li et al., 2020*; *Liu et al., 2019a*; *Liu et al., 2019b*). | Endoscopy dependency | minimally invasive | 9 |
| Esophagus | Esophageal luminal contents | Endoscopic aspiration | There is a difference in the structure of esophageal microbial communities between tissue samples and liquid samples. In esophageal fluid *Firmicutes*, *Bacteroidetes*, *Proteobacteria*, *Actinobacteria*, *Fusobacteria* were the dominant phyla, *Streptococcus*, *Prevotella*, *Veillonella*, *Gemella*, *Rothia*, *Haemophilu* with higher abundance ratios at the genus level (*Jung et al., 2022*). | Endoscopy dependent, invasive | No esophageal mucosal damage. | 8 |
| | | Cytosponge device | (Healthy, Barrett's oesophagus, esophageal adenocarcinoma) The eight most prevalent phyla: *Bacteroidetes*, *Proteobacteria*, *Fusobacteria*, *Actinobacteria*, *Spirochaetes*, SR1, TM7. representative genera: *Veillonella*, *Dialister*, *Selenomonas*, *Megasphaera*, *Granulicatella*, *Oribacterium*, *Catonella*, *Moryella*, *Solobacterium*, *Campylobacter*, *Olsenella*, *Atopobium*, *Actinomyces* (*Elliott et al., 2016*). | The microbial portion sampled from Barrett's esophagus was diluted by a large amount of bacteria in the esophagus, mouth, and stomach. May cause slight pharyngeal bleeding, and the sponge and rope are disconnected when the device is removed. | The examination time is short (5-7 min), safe, acceptable to the subjects, inexpensive, and feasible in secondary healthcare institutions, high microbial DNA yield, which can provide histological data for esophageal inflammatory diseases. | 8 |
| | | Esophageal string test | The microbiota of esophageal biopsy and EST are almost the same. the relative abundance of the main bacterial phyla (*Actinobacteria*, *Bacteroidetes*, *Firmicutes*, *Fusobacteria*, *Proteobacteria*) is similar. The overlap percentage of common bacterial genera is higher than 75%, and the number of genera in biopsy is higher than that in ESTs (*Fillon et al., 2012*). | There is a risk of contamination by adjacent sites. Gagging was noted as the only side effect in some subjects. | Low cost, low risk, large sampling surface area, well tolerated, repeat sampling, provides access to longitudinal samples. | 8 |

**Table 1** (*continued*)

| Organ | Sample source | Sampling | Main sequencing results | Disadvantage | Advantage | NOS |
|---|---|---|---|---|---|---|
| | | Endoscopic cytology brush | At the phylum level, normal gastric mucosa: *Proteobacteria, Firmicutes, Bacteroidetes, Actinobacteria, Verrucomicrobia, Cyanobacteria*; equine glandular gastric disease lesion: *Firmicutes, Proteobacteria, Bacteroidetes, Verrucomicrobia, Actinobacteria, Cyanobacteria* (*Voss et al., 2022*). | Endoscopy dependency | Minimizes tissue damage and host DNA contamination, extensive sampling surface area and relatively low invasiveness, sheathed cytology brush reduces cross-contamination and improves sampling accuracy. | 7 |
| | | Biopsy forceps scraping | The sensitivity of biopsy forceps scraping mucosal rapid urease test was higher than that of tissue biopsy, PCR sensitivity was higher than that of standard biopsy, specificity (99.5%) and overall accuracy (95.3%) were higher than that of standard biopsy samples and gastric juice (*Matsumoto et al., 2016*). | Endoscopy dependency | Large sampling area, easy and safe to operate, does not bleed easily | 7 |
| | Mucosa | Oro-gastric brush | The recovery rate of *Helicobacter pylori* was 100% after storing for 24 h and 72 h before culture (*Graham et al., 2005*) | This study used specially designed brushes | Fast (within 5 min), reliable, minimally invasive, and does not rely on gastroscopy, suitable for sampling in doctors' offices, hospital laboratories, and remote areas. | 8 |
| | Tissue | Endoscopic biopsy | At the phylum level, control: *Firmicutes,* Non-Hp *proteobacteria, Bacteroidetes, Actinobacteria, Fusobacteria*; gastritis: *H. pylori,* Non-Hp *proteobacteria*; early gastric cancer: *H. pylori,* Non-Hp *proteobacteria, Firmicutes*; advanced gastric cancer: *H. pylori, Firmicutes,* Non-Hp *proteobacteria, Bacteroidetes, Fusobacteria, Actinobacteria* (*Sung et al., 2016*). | Most of them are local sampling, with limited sampling area. Those who take anticoagulants or antiplatelet aggregation drugs may not be able to take samples during the medication period and have a high risk of bleeding after sampling. | More effective detection of meaningful gastric microbiota than gastric juice | 7 |
| | | Endoscopic aspiration | At the phylum level, the control: *Actinobacteria,* Non- *H.pylori proteobacteria, Firmicutes, Bacteroidetes*; gastritis: *Firmicutes, Actinobacteria, Bacteroidetes,* Non-*H.pylori proteobacteria, Fusobacteria*; early gastric cancer: *Firmicutes,* Non- *H.pylori proteobacteria, Bacteroidetes, H. pylori*; advanced gastric cancer: *Bacteroidetes,* Non-Hp *proteobacteria, Firmicutes, Fusobacteria, H. pylori* (*Sung et al., 2016*) (*Matsumoto et al., 2016*) | Relying on endoscopy requires more supplies and experienced personnel, as well as a suitable operating environment. | Gastric juice is evenly distributed throughout the entire stomach, reflecting the gastric microenvironment evenly. | 7, 7 |
| Stomach | | Nanofiber capsules | Still in the development and *in vitro* validation stage | Still in the development and *in vitro* validation stage | Minimally invasive, fast, low-cost, suitable for crowd sampling, and can directly sample any part of the digestive tract. | NA |
| | Gastric juice | String-test | Compared with the 13C urea breath test, the diagnostic rate of *H.pylori* infection using string-test qPCR was 95.9%, 93.6% positive, and 100% negative.The eradication rate of *H.pylori* infection under the guidance of drug sensitivity in string-test sample culture was 91.8%, which is higher than empirical treatment (81.3%) (*Han et al., 2023*). | Easy to be contaminated by microorganisms in the nasopharynx and oropharynx, low sensitivity to detecting *H.pylori*, temporary discomfort and vomiting caused by swallowing equipment and removal, interference with daily activities, and difficulty in executing in young children. | Relatively painless, well tolerated, minimally invasive, suitable for children, with high specificity in detecting Helicobacter pylori. | 8 |

Table 1 (*continued*)

| Organ | Sample source | Sampling | Main sequencing results | Disadvantage | Advantage | NOS |
|---|---|---|---|---|---|---|
| | Small bowel aspirates | Gastroduodenoscopy | Main phyla: *Firmicutes* (>50%), *Proteobacteria, Actinobacteria, Fusobacteria, Bacteroidetes, TM7*. *Firmicutes* are mainly *Lactobacillus*, including *Streptococcus, Lactobacillus* and *Botulinum*. *Proteobacteria* are mainly represented by *Neisseria, Pasteurellaceae* and *Enterobacteriaceae* (*Leite et al., 2020*). | Relying on endoscopic examination and requiring sedation, flexible endoscopes cannot undergo thermal disinfection and may form biofilms to confuse microbiological analysis results | Convenience (Routine fluid aspiration during duodenoscopy for better visual field), avoiding oral microbial contamination. | 7 |
| | Tissue | Brisbane Aseptic Biopsy Device, standard biopsy forceps | Main phyla: *Firmicutes* (>60%), *Bacteroidetes, Proteobacteria, Actinobacteria, Fusobacteria, TM7* and *Acidobacteria*; most abundant genera: *Streptococcus, Prevotella, Veillonella, Neisseria, Porphyromonas, Lactobacillus* (*Shanahan et al., 2016*). | Invasive, not suitable for repeated sampling, individuals with intestinal infections at risk of bleeding and infection, limited sampling surface area, and potential sampling bias | Targeted sampling. Biopsy forceps avoid contamination of oral microorganisms through endoscopic channels, while wrapped biopsy forceps can prevent cross contamination in the intestinal cavity and obtain a more representative mucosal associated microbiota. | 7 |
| | Mucosa | Endoscopic cytology brush | (ileal pouch) Mucosal brushings and mucosal biopsies provide comparable results for sampling the mucosa-associated microbiota. The ten most abundant taxa: *Bacteroides, Lachnospiraceae, Clostridium, Enterobacteriaceae, Blautia, Roseburia, Epulopiscium, Peptostreptococcaceae, Acidaminococcus* and *Streptococcus* (*Huse et al., 2014*). Main phyla: *Firmicutes* (>60%), *Bacteroidetes, Proteobacteria, Actinobacteria, Fusobacteria, TM7* and *Acidobacteria*; most abundant genera: *Streptococcus, Prevotella, Veillonella, Neisseria, Porphyromonas, Lactobacillus* (*Shanahan et al., 2016*). | Invasive, endoscopic dependent | Minimal trauma, large sampling surface area, higher bacterial DNA production than biopsy samples, no need for intestinal preparation | 8,7 |
| | | Balloon endoscopy (with sheath cytology brush) | The five most abundant phyla: *Firmicutes, Proteobacteria, Actinobacteria, Bacteroidetes, Fusobacteria*. The most abundant genera are as follows: dodecadactylon (*Veillonella, Streptococcus, Prevotella, Haemophilus, Actinomyces*), jejunum (*Streptococcus, Veillonella, Escherichia, Actinomyces, Haemophilus*), ileum (*Escherichia, Haemophilus, Streptococcus, Bacteroides, Veillonella*), terminal ileum (*Escherichia, Bacteroides, Haemophilus, Streptococcus, Clostridium* cluster XIVa) (*Nagasue et al., 2022*). | It is invasive, expensive, and the achievable depth of the small intestine depends on the physical condition of the subject and the technical level of the endoscopist, with high technical requirements for the operator | The entire small intestine can be sampled, and the mucosa deep in the small intestine can be collected under physiological conditions. | 8 |
| | Duodenal intraluminal fluid | 3D sampling capsule | It is still in the development and *in vitro* validation phase | It is still in the development and *in vitro* validation phase | Low cost, non-toxic, harmless, with high comfort and no need for clinical testing, capable of collecting samples at different target locations throughout the gastrointestinal tract | NA |

**Table 1** (*continued*)

| Organ | Sample source | Sampling | Main sequencing results | Disadvantage | Advantage | NOS |
|---|---|---|---|---|---|---|
| Small intestine | Mucosal | Operation (Wipe with a swab) | The most frequent bacteria were *Streptococcus salivarius/vestibularis, S. parasanguinis, S. mitis/oralis, Rothia mucilaginosa, Actinomyces odontolyticus, Haemophilus parainfluenzae, Neisseria flavescens/subflava* and *Neisseria parahaemolyticus* (*Villmones et al., 2022*). | There are challenges involved ethical and patient safety. | Not contaminated by other parts of the gastrointestinal tract. | 8 |
| | Effluent samples from ileostomists | Postoperative flow through the stoma after small intestinal fistulization | *Bacilli* (*Streptococcus sp.*), *Clostridium* clusters XIVa (several genera) have a high abundance in the ileostoma effluent (*Zoetendal et al., 2012*) | Risk of contamination by adjacent skin and external environment | Allow non-invasive repeated sampling of patients undergoing ileostomy. | 7 |
| | Feces | Direct collection of feces and feces | The main bacterial phyla in the feces of healthy individuals include *Firmicutes, Bacteroidetes,* and *Proteobacteria* (*Radhakrishnan et al., 2023*). At the genus level, stool samples had higher relative abundances of members of the *Akkermansia, Bacteroides, Enterococcus,* and *Parabacteroides* taxa, which are considered typical members of the gut microbiome in critically ill patients (*Fair et al., 2019*). The most common bacterial pathogens in fecal culture include *Shigella, Salmonella*, Shiga toxin-producing *E. coli*, and *Campylobacter* (*Jean et al., 2019*) | Relying on patients to collect samples themselves and unable to provide samples on demand in a timely manner.It is often difficult to obtain specimens in the intensive care unit, patients are unwilling to handle feces, and feel embarrassed about transporting feces | Non invasive, sufficient sample size. It is still Common method for studying gut microbiome | 7 |
| | | FOBT card | Compared with RNA later samples, the abundance of *Bacteroidetes* in fecal samples collected by FOBT cards is significantly lower, but the abundance of *Actinobacteria* and *Firmicutes* is higher (*Wu et al., 2021*). | The amount of fecal samples collected is relatively small (only a small amount of feces will be applied to the FOBT card) | It has acceptability and practicability, and is cost-effective for large-scale cohort study. | 8 |
| | | Rectal swab | The swab method has similar microbial community results with feces. the phylum *Proteobacteria* and the genus *WAL-1855D* (a Sporobacterium) is enriched in swab samples (*Short et al., 2021*) | There is a significant amount of human DNA. It may be contaminated by perianal skin or bacterial communities in the female urogenital system. | Sampling is simple and convenient, non-invasive, easy to manage and transport samples, frozen, and highly accepted by participants. It can be the preferred specimen for outbreak investigations in public health environments. Maintaining detection sensitivity at 4 °C can replace feces for studying intestinal microbiota, especially in elderly and critically ill patients with constipation. | 8 |
| | | Glove tip after rectal digital examination | Glove-tip collection is similar to swab collection techniques and is often similar to household fecal collection. *Oscillospira* is enriched in glove tip samples (*Short et al., 2021*) | The materials obtained are limited, providing sufficient samples for 16SrRNA evaluation, but may not be suitable for metabolomics analysis. | The technique is simple, requires no advance preparation by the participant, and the specimen is easy to transport, avoiding the need for a mucosal biopsy to collect a mucosal specimen. | 8 |

**Table 1** (*continued*)

| Organ | Sample source | Sampling | Main sequencing results | Disadvantage | Advantage | NOS |
|---|---|---|---|---|---|---|
| Colon and rectum | Tissue | Endoscopic biopsy | The main members of the gut microbiota include *Clostridium spp*, *Bifidobacterium spp*, *Bacteroides spp*, *Lactobacillus spp* and *E. coli* (*Araújo Pérez et al., 2012*) | Relying on endoscopy, testing is expensive and time-consuming. Intestinal preparation may affect the gut microbiota. it is difficult to obtain a large number of specimens, and is not suitable for healthy subjects. | Helping to analyze mucosal associated microbiota, routine tissue biopsy is convenient for those undergoing endoscopic examination. | 8 |
| | Mucosa of cavity wall | Mucosal brush | At the phylum level, the abundance of *Actinobacteria* is more abundant at the endoscopic brush sample level, while the abundance of *Bacteroidetes* is less rich. In the class levels, *Actinobacteria* (*Bifidobacteriale*) and *Bacilli* (*Lactobacillales*) tended to be richer in brushing samples, while *Bacteroidia (Bacteroides)* is less rich. At the genus level, the abundance *Veillonella, Bulleidia* and *Corynebacterium* in the ascending colon and *Lactobacillus* and *Corynebacterium* in the sigmoid colon significantly increased in brushing samples (*Matsumoto et al., 2019*). | Relying on endoscopic, examination reduces biodiversity. | Controllable sampling area, providing a large number of biological specimens | 7 |
| | Intestinal lavage fluid | Endoscopic aspiration | The dominant bacteria in the lavage samples were *Firmicutes, Bacteroidetes, Proteobacteria, Actinomycetes* (*Watt et al., 2016*). The main bacterial phyla *Firmicutes, Bacteroides, Proteobacteria, Actinobacteria, Fusobacteria* (*Kwon et al., 2021*) | Relying on endoscopy, detection is time-consuming and invasive, possibly including the gut microbiota and mucosal microbiota within the cavity | Compared with biopsy samples, microbial diversity and uniformity are similar, and colon lavage samples have significantly higher DNA content and higher microbial counts | 8 |
| | Bile | ERCP | The most dominant phyla were *Proteobacteria* (61.7%), *Firmicutes* (25.1%), *Bacteroidetes* (5%), *Fusobacteria* (4.6%) and *Actinomycetes*(2.6%).The content of *Enterococcus* in bile of patients with brown pigment stone was rich (*Kim et al., 2021*). *Bacteroides fragilis* is significantly associated with cholangitis (*Effenberger et al., 2023*). | It is invasive and may disrupt the normal anatomy and functional barrier of the biliary tract, increasing the risk of infection in the ascending biliary tract. Residual bacteria in the oral cavity and endoscopy can be transferred to the biliary tract, causing contamination of the biliary system. | Minimally invasive | 7, 8 |
| | | Cholecystectomy | The main bacterial phyla of bile include *Firmicutes, Bacteroidetes, Actinobacteria, Proteobacteria*. In patients with cholelithiasis members of the families *Bacteroidaceae*, *Prevotellaceae*, *Porphyromonadaceae*, *Veillonellaceae* were more frequently detected (*Molinero et al., 2019*) | Not suitable for healthy individuals | Suitable for patients with indications for cholecystectomy | 7 |

**Table 1** (*continued*)

| Organ | Sample source | Sampling | Main sequencing results | Disadvantage | Advantage | NOS |
|---|---|---|---|---|---|---|
| Liver, gallbladder, and bile duct | HCC organization | Resection | Dominant phylum: *Proteobacteria, Firmicutes, Actinobacteriota, Bacteroidota*. Dominant genera: *Aliidiomarina, Halomonas, Dietzia,* and *Achromobacter*. At the phylum level, *Actinobacteriota* and *Verrucomicrobiota* genera had significantly lower abundance in HCC tissues than in paraneoplastic tissue. At the genus level, the abundances of the genera *Dietzia, Faecalibacterium, Megamonas, Hydrogenophaga, Agathobacter, Chryseobacterium,* and *Ruminococcaceae* were significantly lower, while the abundances of *Neisseria, Clostridia_UCG-014, Fusobacterium,* and *Lactobacillus* were significantly higher in HCC tissues than in paracancerous Tissues (*He et al., 2023*). | Not suitable for healthy individuals and patients with unresectable tumors | Suitable for those with resectable HCC | 8 |
|  | ICC | Resection | Bacteria with high abundance included *Burkholderiales, Pseudomonadales, Xanthomonadales, Bacillales, Clostridiales,* and *Sphingomonadales*. Using the bacterial culture experiments, *Staphylococcus* capitis in fresh tumor tissue of ICC was found (*Chai et al., 2023*). | Not suitable for healthy individuals and patients with unresectable tumors | Suitable for those with resectable ICC | 8 |
| Pancreas | Pancreatic cancer tissue | Resection | Major bacterial phyla: *Proteobacteria* (45.9%), *Firmicutes* (35.6%), *Bacteroidetes* (9.5%), *Fusobacteria* (4.3%), and *Actinomycetes* (3.9%) (*Del Castillo et al., 2020*). | Not suitable for healthy people and unresectable patients with pancreatic cancer | Suitable for patients with indications for pancreatectomy | 8 |
|  |  | EUS-FNA | Main bacterial phylum: *Proteobacteria, Firmicutes, Bacteroidetes, Actinobacteria, Fusobacteria*. There was no significant difference in tumor bacterial diversity and composition at the bacterial phylum level between resectable and unresectable pancreatic cancers. *Delftia* is higher in resectable pancreatic cancer (*Nakano et al., 2022*). | The amount of tissue obtained may not reflect the bacterial diversity and composition of the entire pancreas. Pancreatic cancer tissue was collected by EUS-FNA from the gastric or duodenum wall, which may contaminate the samples. | The smallest pancreatic cancer tissue can be used for pancreatic cancer microbiome analysis | 7 |
|  |  | EUS-FNB | There was no significant difference in alpha diversity, beta diversity or taxonomic profiles between EUS-FNB and surgical resection Samples (*Masi et al., 2021*) | The sampling process is invasive and carries a risk of contamination. | Sufficient fresh cancer tissue can be collected, suitable for resectable and non resectable pancreatic cancer patients | 7 |
|  | Pancreatic cyst fluid | EUS-FNA | *Bacteroides spp., Escherichia/Shigella spp., Acidaminococcus spp.* which were predominant in pancreatic cyst fluids, while also a substantial *Staphylococcus spp.* and *Fusobacterium spp.* component was detected (*Li et al., 2017*) | The sampling process is invasive and carries a risk of contamination | Minimally invasive | 8 |

Notes.

OTUs, operational taxonomic units; DNA, deoxyribonucleic acid; EST, esophageal string test; PCR, polymerase chain reaction; ERCP, retrograde cholangiopancreatography; HCC, hepatocellular carcinoma; ICC, intrahepatic cholangiocarcinoma; EUS-FNA, endoscopic ultrasound-guided fine needle aspiration; EUS-FNB, endoscopic ultrasound-guided fine needle biopsy; NA, not available.

**Institutional abbreviations**

| | |
|---|---|
| BABD | brisbane aseptic biopsy device |
| ERCP | endoscopic retrograde cholangiopancreatography |
| PTCD | percutaneous transhepatic cholangiography drainage |
| FFPE | formalin-fixed paraffin-embedded |
| OTUs | operational taxonomic units |
| DNA | deoxyribonucleic acid |
| EST | esophageal string test |
| PCR | polymerase chain reaction |
| ERCP | retrograde cholangiopancreatography |
| HCC | hepatocellular carcinoma |
| ICC | intrahepatic cholangiocarcinoma |
| EUS-FNA | endoscopic ultrasound-guided fine needle aspiration |
| EUS-FNB | endoscopic ultrasound-guided fine needle biopsy |

## ACKNOWLEDGEMENTS

We sincerely thank the researchers included in this review for their outstanding contributions.

### Funding

This study was funded by the Health Commission of Hubei Province scientific research project (grant number: WJ2023Q022), the Shiyan City Science and Technology Bureau Guiding Research Project (grant number: 21Y19). The funders had no role in study design, data collection and analysis, decision to publish, or preparation of the manuscript.

### Grant Disclosures

The following grant information was disclosed by the authors:
Health Commission of Hubei Province scientific research project: WJ2023Q022.
Shiyan City Science and Technology Bureau Guiding Research Project: 21Y19.

### Competing Interests

The authors declare there are no competing interests.

### Author Contributions

- Xiaobo Liu conceived and designed the experiments, performed the experiments, prepared figures and/or tables, authored or reviewed drafts of the article, and approved the final draft.
- Xia Cheng conceived and designed the experiments, performed the experiments, analyzed the data, prepared figures and/or tables, authored or reviewed drafts of the article, and approved the final draft.
- Ziye Gao conceived and designed the experiments, performed the experiments, prepared figures and/or tables, and approved the final draft.

- Jun Pan analyzed the data, prepared figures and/or tables, and approved the final draft.
- Shizhen Luo performed the experiments, analyzed the data, prepared figures and/or tables, and approved the final draft.
- Pei Liu analyzed the data, prepared figures and/or tables, authored or reviewed drafts of the article, and approved the final draft.
- Hui Wen conceived and designed the experiments, performed the experiments, prepared figures and/or tables, authored or reviewed drafts of the article, and approved the final draft.
- Shu Jin conceived and designed the experiments, authored or reviewed drafts of the article, and approved the final draft.

## Data Availability

This is a literature review.

## Supplemental Information

Supplemental information for this article can be found online at http://dx.doi.org/10.7717/peerj.19810#supplemental-information.

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
