# Peer review of "Sampling strategies for digestive system flora studies: current research and perspectives"

_PeerJ, doi:10.7717/peerj.19810_

## Round 0.1 · original submission · Major Revisions

Dear authors,

As outlined by the reviewers your article deals with an interesting important topic. However, the reviewers encountered several issues in the manuscript which must be fixed before your manuscript can be considered for publication.

Kind regards,
Elisabeth Grohmann

·

Basic reporting

The authors of the present submitted manuscript have systematically reviewed biological samples and collection methods used in digestive system microbiota research to inform future strategies. The samples include saliva, endoscopic biopsies, luminal mucosa, luminal fluid, feces, bile, and surgically excised tissues. However, certain aspects require further clarification.

Introduction:
The introduction should provide a more detailed explanation of the sampling methods, as this is the primary focus of the article. This will better prepare the audience to comprehend the subsequent material.

Experimental design

Method:
1. In the Method section, begin by outlining the study design section, including its goals. Additionally, mention the utilization of the PRISMA flowchart and provide the PROSPERO registration code ID.
2. When selecting a search strategy, it is advisable to explore Scopus and Cochrane. Also, expand your keyword selection to ensure comprehensive coverage of potential articles, such as including "biliary." Moreover, clearly outline your sampling method and integrate it with your microbiome-related keywords during the search process.
3. Your article appears to lack a quality assessment section despite concerns raised about research quality. How do you assess the studies included in your article? Do you utilize methods such as the Newcastle-Ottawa Scale?
4. The PRISMA chart should include unrelated articles, such as reviews and editorials, and indicate how many are in a language other than English. Additionally, please provide the rationale for excluding specific articles and summarize accurate statistics about the excluded studies. Your PRISMA chart requires improvement.

Validity of the findings

Results:
When presenting your findings, it would be beneficial to include a section with general information about your article. This section could categorize relevant information about the included studies. Indeed, it would also be helpful to summarize this information in a separate table, especially since this is a systematic review.

Discussion:
In the Discussion section, avoid resembling a conclusion. Instead, discuss the methods mentioned, outlining their advantages and disadvantages with appropriate references to support your points. It would be better to improve your discussion.

Reviewer 2 ·

Basic reporting

One of the most significant hurdles in laboratory medicine is the implementation of effective sampling strategies. I'm grateful for the opportunity to review this comprehensive paper.
I believe that this manuscript offers a thorough review of the methods, highlighting their benefits and limitations.
in my opinion, the manuscript will be publishable with some minor revisions:

1- A brief explanation of the mechanism and procedure of EST and Cytosponge techniques is recommended in the esophagus section.
2- line 58, resembles substitute for resemble
3- line 60, add a space after time
4- line 66, check the space between conversely and diseases
5- line 67, add a space after microbiota
6- line 68, add a space after microbiotas
7- line 90, add a space between information and that
8- line 109, substitute articles for article
9- line 117, substitute have for has
10- line 125, substitute methods for method
11- line 152, add a space after the reference and considering
12- line 287, add a space between the two references
13- line 205, substitute were for was
14- line 242, add a space after the reference and conducted
15- line 253, substitute suitable for is suitable
16- line 281, add a space after the reference and from
17- line 286, substitute Brisbane for brisbane
18- line 311, add a space between regions and the reference
19- line 315, substitute However for however
20- line 319, substitute were for was
21- line 329, substitute offer for offers
22- line 345, add a space between limited and the reference
23- line 356, substitute is, provides, yields for are, provide, yield
24- line 394, add a space between the reference and team
25- line 421, substitute individuals for individual

Experimental design

no further comment

Validity of the findings

no further comment

Reviewer 3 ·

Basic reporting

Introduction

Quantification of Microbiota:
The statement "the digestive tract microbiota constitutes approximately 70% of the total microbiota" (line 56) is quite broad. Clarify whether this percentage refers to the number of microorganisms, biomass, or another metric.

Specificity of Sources and Sampling Methods:
The mention of various sources for microbiota samples (lines 73-75) is good, but the text should elaborate on how these sources differ in microbial composition and what impact these differences might have on research outcomes. The text could also benefit from discussing specific advantages and limitations of each sampling method.

Experimental design

no comment

Validity of the findings

Results

Depth and Scope:
The section (337-340 lines) does not adequately cover the depth and scope of microbial diversity within different regions of the colorectum. More detailed discussion on how microbial populations differ across the colon and rectum would be valuable.

Practical Recommendations:
The text should provide practical recommendations for researchers on which sampling methods to use under specific circumstances. This guidance would be useful for those planning colorectal microbiota studies.

Additional comments

Discussion and Outlook

Bioinformatics Tools:
The mention of bioinformatics tools like Deconam is brief and lacks detail. A more thorough discussion on the role of bioinformatics in microbiota research, including specific tools and their applications, would be beneficial.

Impact on Clinical Practice:
The potential impact of these findings on clinical practice, such as how understanding microbiota can inform treatment strategies for liver, biliary tract, and pancreatic diseases, is not sufficiently addressed. Including a section on clinical implications would add significant value.

Reviewer 4 ·

Basic reporting

no comment.

Experimental design

no comment.

Validity of the findings

1. It needs to be described in more detail in the Discussion and Outlook part. Especially in the outlook part, in addition to some obvious summaries and descriptions, the author's more unique personal views should be explained, which will help to convey the author's thoughts more clearly.

Additional comments

1. Table 1 is very rich, which is very important to guide readers' understanding, so the author needs to briefly introduce the contents of Table 1 to readers in the text.
2. The author needs to check the consistency of sentences in the whole text, such as lines 257 to 260 and lines 213 to 215 of the manuscript.-The current logic is hard to understand. Please recap it.
3. In the RESULTS section, listing commonly used biological samples and sampling techniques in the first paragraph of each section will help to improve the hierarchy of the article.

---

## Round 0.2 · accepted · Accept

Dear authors, Your manuscript has considerably improved by your thorough revision and is now acceptable for publication in the journal. However, I kindly request you to carefully revise spelling throughout the manuscript and check spaces after the insertion of brackets, especially when citing references. Please submit this revised manuscript to the journal for publication. Thank you

Best regards,
Elisabeth Grohmann

Reviewer 2 ·

Basic reporting

The revised manuscript has significantly improved. The authors have responded clearly and comprehensively to the reviewer comments. I find no remaining issues, and I support publication.

Experimental design

/

Validity of the findings

yes